Evaluation of Leymus chinensis quality using near-infrared reflectance spectroscopy with three different statistical analyses

Chen Jishan 1 2
Zhu Ruifen 2
Xu Ruixuan 1
Zhang Wenjun 1
Shen Yue 1
Zhang Yingjun 1 zhangyj@cau.edu.cn
1 Department of Grassland Science, China Agricultural University , Beijing , China
2 Heilongjiang Academy of Agricultural Science, Institute of Pratacultural Science , Harbin , China
Goswami Chandan
Electronic publication date: 2015 Dec 3
Publication date: 2015
Volume: 3
Electronic Location ID: e1416
Received 2015 Sep 2; Accepted 2015 Oct 29
Copyright: © 2015 Chen et al.
Copyright year: 2015
Copyright holder: Chen et al.
License: This is an open access article distributed under the terms of the Creative Commons Attribution License, which permits unrestricted use, distribution, reproduction and adaptation in any medium and for any purpose provided that it is properly attributed. For attribution, the original author(s), title, publication source (PeerJ) and either DOI or URL of the article must be cited.
License URL: https://creativecommons.org/licenses/by/4.0/

Keywords: Near infrared spectroscopy, Chemical quality, Sheepgrass (Leymus chinensis), Root mean squares error of calibration (RMSEC), Root mean squares error of prediction (RMSEP)

Funding: This study was funded by the China Agriculture Research System (CARS-35). The funders had no role in study design, data collection and analysis, decision to publish, or preparation of the manuscript.

==============================
Due to a boom in the dairy industry in Northeast China, the hay industry has been developing rapidly. Thus, it is very important to evaluate the hay quality with a rapid and accurate method. In this research, a novel technique that combines near infrared spectroscopy (NIRs) with three different statistical analyses (MLR, PCR and PLS) was used to predict the chemical quality of sheepgrass (Leymus chinensis) in Heilongjiang Province, China including the concentrations of crude protein (CP), acid detergent fiber (ADF), and neutral detergent fiber (NDF). Firstly, the linear partial least squares regression (PLS) was performed on the spectra and the predictions were compared to those with laboratory-based recorded spectra. Then, the MLR evaluation method for CP has a potential to be used for industry requirements, as it needs less sophisticated and cheaper instrumentation using only a few wavelengths. Results show that in terms of CP, ADF and NDF, (i) the prediction accuracy in terms of CP, ADF and NDF using PLS was obviously improved compared to the PCR algorithm, and comparable or even better than results generated using the MLR algorithm; (ii) the predictions were worse compared to laboratory-based spectra with the MLR algorithmin, and poor predictions were obtained (R2, 0.62, RPD, 0.9) using MLR in terms of NDF; (iii) a satisfactory accuracy with R2 and RPD by PLS method of 0.91, 3.2 for CP, 0.89, 3.1 for ADF and 0.88, 3.0 for NDF, respectively, was obtained. Our results highlight the use of the combined NIRs-PLS method could be applied as a valuable technique to rapidly and accurately evaluate the quality of sheepgrass hay.

Introduction

As an important perennial forage grass across the Eurasian Steppe, sheepgrass (Leymus chinensis (Trin.) Tzvel.) is known for its adaptability to various environmental conditions (Chen et al., 2013), high palatability, rich in nutrients and all kinds of livestock feeding (Liu & Qi, 2004; Yang, Liu & Zhang, 1995). Leymus chinensis has tender stems and leaves, and high forage yield without irrigation about 3,000 to 4,500 kg/hm2, and the yield with irrigation reaches 6,000 kg/ hm2 in the Northeastern Plain and east of the Inner Mongolian Plateau (Chen, 2001), and it may contribute to good balanced diet for cows fed with sheepgrass hay on milk production and composition (Yan et al., 2011).

In the last 20 years, the population of Northeast China has increased considerably, resulting in steeply increase in numbers of livestock. Hence, sheepgrass in this region is regarded as an important productive grass for the hay industry, which is developing rapidly due to the prosperous status in the dairy industry. A great deal of animal farms with approximately 20,000 sheeps or 10,000 cattles have been or are being established in Northeast China, resulting in an urgent need for forage, including commercial forage and natural herbage. Annually, more than 30 companies produce hay over 117.19 million tons of natural herbage from sheepgrass in Heilongjiang Province, China (http://www.caaa.cn/association/grass/). Consequently, a large number of sheepgrass hay produced by personal goes into the market and become a commodity. The difference between commercial forage and natural herbage dominanted by sheepgrass is that the former is attached a detailed trademark to the hay productors entered into the market, while the latter is absent from the quality indices of natural herbage dominanted by sheepgrass during selling.

Nevertheless, sheepgrass is one of important types of hay in the everyday life of an animal, and the final purchase decision by the buyer is often according to the feeding value, which is well related with terms of chemical and biological components. Because of the impact of factors such maturity period at harvest, botanical components, and cutting techniques on the production process in different climate, the principal difference between natural herbage and commercial forage is the complexity of raw materials of the former (dominanted by sheepgrass), and measurements of internal quality indices of sheepgrass hay cost time and are destructive. Therefore, the establishment of a rapid and accurate method with nondestructive for evaluate chemical qualities of sheepgrass hay is extremely important to the hay industry before selling.

Assessing the quality of natural herbage dominanted by sheepgrass is very important for high quality forage with various parameters, namely the concentrations of crude protein (CP), acid detergent fiber (ADF) and neutral detergent fiber (NDF), which are commonly used to assess the forage quality. However, all analytical procedures are time-consuming or expensive when a great quantity of samples are involved (Yu et al., 2010; Yu et al., 2011; Albrecht et al., 2009). Near infrared reflectance spectroscopy (NIRs) with wavelength range of 750–2,500 nm has much superiority over chemical analyses in a laboratory, for example its ease of sample preparation, rapid spectrum acquisition, non-destructive nature of the analysis, and the portability of the technology (Albrecht et al., 2009). Many works have proved that NIRs is widely used to assess the quality of forage dominanted by alfalfa, oats, silage corn, ryegrass and so on (Nie, Han & Yu, 2007; Ding et al., 2009; En et al., 2009; Liu, Zhang & Xu, 2014; Hu et al., 2014; Bai, Chen & Dong, 2004; Chen, Rong & Han, 2007), and it is also being implemented in sheepgrass (Shi & Zhang, 2011). However, it remains unclear whether NIRs is applied to search the valuable information about NIRs prediction models and rapidly and accurately assess the quality of the hay dominanted by sheepgrass for CP, ADF and NDF.

In addition, the spectrum of sheepgrass hay obviously shows some peaks and valleys in the wavelengths from 950 to 1,650 nm, which includes hidden information of different components, and this does not mean that some useful information cannot be extracted in other wavelengths (Liu, Chen & Ouyang, 2008). To predict and determine the quality parameters of sheepgrass hay, multivariate statistics analysis techniques, such as multiple-linear regression (MLR), principal component regression (PCR) and partial least squares regression (PLS), are applied to establish the prediction models by analyzing correlations between measured chemical values and the spectrum measurements of sheepgrass hay in this study.

The objective of this study was to search the prediction models of NIRs to determine the essential quality indices of sheepgrass hay. A total of 203 samples of sheepgrass hay were collected from 37 sampling sites distributed throughout Heilongjiang Province, China. Our purpose of this study were to evaluate the performance of NIRs in measuring CP, ADF, and NDF of sheepgrass hay, and compare the prediction potentiality of different methods (MLR, PCR and PLS) for rapidly and accurately evaluate the quality of sheepgrass hay.

Materials and Methods

Sample collection and pretreatment

The sampling sites were from the west to east across the grassland in Heilongjiang Province of Northeast China. A total of 203 samples of sheepgrass hay were randomly collected from sheepgrass fields of the hay factories in 2013, with latitudes ranging from 44.475°N to 51.728°N, longitudes from 123.209°E to 132.944°E. The sites were chosen to be representative of sheepgrass production fields and contain a range of soils and climate are described in Appendix S1. These locations produce approximately 117.19 million tons of sheepgrass hay per year. All samples of sheepgrass hay were sampled at blooming stages, identified and collected before being clipping and packaging. The collected samples (not involve endangered or protected species), a total sample size of 203 from 21 sampling sites distributing over 8 regions (not privately owned or protected in any way) in Heilongjiang Province (see Fig. 1 and Appendix S1), were then forwarded to the lab and stored at 4 °C for further analyses.

Figure 1 Summary of sampling sites distribution in Heilongjiang Province, China.

To be representative, each sample consisted of one quarter square meter clipped at 4 cm, transported to the lab, oven dried (65 °C, 48 h), ground (1 mm sieve), and mixed. Meanwhile, the mixed samples were divided by the quartile method; half for duplicate chemical analyses and half for the near infrared reflectance spectra.

In fact, all samples (203) were used to evaluate and develop NIRs models in this study. To ensure the adaptability of the calibration models, some samples (51) were used for prediction set and the rest of samples (152) for calibration set. All tasks including spectral measure and chemical analyses were finished on the same day or the next day.

Chemical properties analyses

The first experiment was designed to develop a database to evaluate relationships between the quality indices of sheepgrass hay and NIRs measurements. In order to accomplish this goal, three characteristics were measured at blooming stages of growth. These included CP, ADF, and NDF, which were usually regarded as the principal forage quality paremeters, serving as the primary nutrition source in the diet of dairy cattle (Yan et al., 2011; Mertens, 1992). All chemical analyses were conducted in duplicate using analytical grade chemicals are described in Appendix S2. The CP was quantified using the Kjeldahl procedure for nitrogen (N) determination (%CP =%N × 6.25) by the methods of Association of Official Analytical Chemists-AOAC (AOAC, 1990). The ADF and NDF were conducted by using the method described by Van Soest, Robertson & Lewis (1991).

Spectroscopic measurement

By using an near infrared spectrometer (DA7200, Perten Corporation, Hägersten, Sweden) to collect for the spectra of samples, the NIR spectra were recorded at 5 nm intervals from 950 to 1,650 nm in Appendix S2. Approximately 50 g of dried sample was scanned in a 7.5 cm diameter sample cell with a quartz window in the room temperature maintained at 25 °C. With the working parameters of DA7200 software, samples were scanned from 950 to 1,650 nm in reflectance mode (R mode). Data form was converted to the absorbance (the logarithms of the reflectance reciprocal). In order to eliminate an error caused by loading the sample, each sample were repeated to scan three times and each time with scanning background light three times. The average spectral scanning were as for the final spectra of samples deposited in the computer for calculating using the Grams32 software (Perten Corporation, Hägersten, Sweden).

Statistical analysis

To obtain linear correlations of the NIRs with the chemical values, spectra were expressed as the absorbance A (A=log1/R). The NIRs were conducted by Grams32 software and Microsoft Office Excel 2003. In performing measurements with the NIR System DA7200 spectrometer, a number of data were generated for each sample and it is obvious that some data reduction method is needed to facilitate data interpretation. Thus, there is a need for data processing methods that transform the measured spectral data into the sample properties of interest. For producing such models that determine the equations describing the relationship between spectral data and chemical values, three different methods (MLR, PCR and PLS) were used. MLR uses a multiterm linear polynomial to describe this relationship using only several spectral data measured as “characteristic” wavelengths (by the way of correlation spectrum). With PCR, all spectral data are used, reducing the dimensionality of the data set by looking for orthogonal directions in spectral data space along which the variance of the data set is maximised. These directions are called principal components (PC). Thus, the first principal component (PC 1) is determined as the direction in spectral data space that corresponds to the largest variation in the data set. The second principal component (PC 2) is calculated as the direction perpendicular to PC 1 along which the remaining variation in the data set is the largest. This procedure is repeated until no variation is left in the data set. PLS uses not only the spectral information in the data set but also incorporates information about sample properties, e.g., concentrations to determine the most useful orthogonal directions in spectral space.

An ideal equation would have a coefficient of determination (R2) of 1.0 and root mean squared error (RMSE). Approximately 75% of the data were used in creating the equation, while the remaining 25% were randomly removed for validation. Validation procedures for the created equations were the same as described above, using the remaining 25% of the data. In this study, the calibration set (152 samples) and the validation set (51 samples) were randomly divided by the method applied by several publications (Miriam et al., 2011; Peltre et al., 2011; Hayes, 2012). To restrain invariable background signals and to improve the visual resolution, the Savitsky-Golay 2nd order derivations were also applied in this study. The selection of the models developed was largely dependent upon determination coefficient for the calibration set (R2) and validation set (r2), root mean squares error of calibration (RMSEC), the RPD (ratio of standard error of performance to standard deviation) and root mean squares error of prediction (RMSEP) used to evaluate the practicability of NIRs for determining to the quality indicators of sheepgrass hay.

The formula for RMSEC and RMSEP are described briefly below: RMSEC=1nc∑i=1ncyi¯−yi2

RMSEP=1np∑i=1npyi¯−yi2

where yi¯ is the predicted value of the ith observation; yi is the measured value of the i-th observation; nc is the number of observations in calibration set; np is the number of observations in validation set.

Results

Chemical value

The obtained data (chemical value) were calculated and analyzed by the min, max, average, and standard deviation (SD), while the results of the calibration sets and validation sets were summarized in Table 1. These data sets indicated a wide variability in the chemical indices. For all the total sheepgrass sampled, the CP of 203 samples ranged from 6.2% to 14.33% (SD = 1.24), while the ADF ranged from 35.13% to 42.34% (SD = 2.33) and NDF from 50.71% to 71.08% (SD = 2.67). The datum of CP and NDF were nearly normally distributed around the average (10.54% and 60.89%) but the distribution of ADF measurements was skewed to the right, with the right tail stretching further than the left tail (mean = 38.74%).

Table 1 Summary statistics calibration and prediction sets for CP, ADF and NDF of sheepgrass hay by laboratory reference methods (DM%).

Parameters	Data set	N	Min	Max	Mean	SD	
CP (%)	Total samples	201	6.20	14.33	10.54	1.24	
	Calibration set	150 (2)	6.25	14.33	10.50	1.22	
	Prediction set	51	6.15	14.32	10.57	1.25	
ADF (%)	Total samples	195	35.13	42.34	38.74	2.33	
	Calibration set	144 (8)	35.62	42.30	38.96	2.35	
	Prediction set	51	34.63	42.38	38.51	2.30	
NDF (%)	Total samples	202	50.71	71.08	60.89	2.67	
	Calibration set	151(1)	50.20	70.66	60.43	2.69	
	Prediction set	51	51.21	71.50	61.34	2.68	
Notes.

CP crude proteinelectrical

NDF neutral detergent fiber

ADF acid detergent fiber

SD standard deviation

DM dry matter

The bracketed numbers are outliers during the calibration process.

This wide heteromorphosis in the quality indices of the sheepgrass hay was beneficial to successfully establish a relationship between the NIRs and the quality indices (Yu et al., 2010). Outliers in the collected sample were considered and thus excluded during the process of calibration. The elimination of outliers was based on the criterion that if mahalanobis distance for a sample was 3 SD or more (Yang, Han & Fan, 2006). For the value of chemical indices in this study, 2 outliers were removed for CP, 8 for ADF, and 1 for NDF (Table 1).

NIR spectra

Some peaks and valleys were obviously shown in the spectra (Fig. 2), which represent the characteristics of sheepgrass hay including hidden information of different components and its quantities. All the near infrared reflectance spectra of the collected sample were separated the two groups of spectra with different slopes, one group displayed an obvious absorbance peak at wavelengths of nearly 1, 100 nm, another group showed significant spectral peak was at approximately 1, 450 nm (Fig. 2). From the NIRs of the total sheepgrass samples (N = 203), the absorbance peaks were very overlapped, primary reason that the spectrum includes combinations and overtones of vibration such as stretching and bending of hydrogen-bearing functional groups such as –CH, –OH, and –NH (Saeys, Mouazen & Ramon, 2005). Meanwhile, other interference information affects the near infrared reflectance spectra (Meissl et al., 2008). Therefore, the quantitative evaluation are of difficulty through NIRs alone. Multivariate methods might be necessary to analyse the response of quality of sheepgrass from spectral characteristics with the support of chemometric methods, e.g., PLS, PCR and MLR analysis (Meissl et al., 2008; Saeys, Mouazen & Ramon, 2005).

Figure 2 Spectra of NIR of a total 203 sheepgrass samples.

Multivariate calibration analysis

For quantitative evaluation methods multiple linear regression (MLR), principal component regression (PCR) and partial least squares regression (PLS) were used to for data modelling and for predicting the investigated chemical constituents. For model validation, full cross-validation was used; each case was predicted by the model derived from all other remaining spectra. The results of full cross-validation is the average of the standard error of prediction values produced during each cross-validation step. The optimum number of Orthogonal factors (PC) for the models was obtained by using the leave-one-out cross validation technique for the calibration set. The recorded log (1/R) spectra were smoothed and transformed to second derivative before the analysis. The optimum models were achieved in Table 2 for the spectral processing method of CP, ADF, and NDF. PLS and PCR methods are based on the regression of the full spectra while MLR is based on discrete parts of the spectra (2 wavelengths) in this study. Table 2 showed that the comparison of the accuracy of the calibration results achieved by near infrared spectroscopy using PLS, PCR, and MLR methods, respectively, for evaluation the chemical indices of samples selected.

Table 2 Main parameters of three best calibrations for three different methods.

Parameters	PC	Data format	Filtering method	Filtering parameter	
CP (%)	8	Second derivative	Savitzky-Golay filter	3,2	
ADF (%)	7	Second derivative	Norris derivative filter	5,2	
NDF (%)	10	Second derivative	Norris derivative filter	5,2	
Notes.

For Sacitzky-Golay filter, the paramerers include data points and polynomial order.

For Norris derivative filter, it means segment length and gap between segment.

PC number of principal component

The results of the calibration procedures are presented in Table 3. It can be seen in the table that the smallest value for prediction error (RMSEC) with the highest coefficient of determination for the calibration set (R2) was provided by the PLS method. The results were less accurate for CP, NDF and ADF provided by the PCR method. The MLR had the worst prediction. Therefore, predictions were moderately successful for CP, NDF, and ADF was provided by the PLS method with second derivative of log(1/R) spectra.

Table 3 Comparison of the accuracy of the calibration results (n = 152) achieved by NIRS using three different methods for evaluation.

	PLS	PCR	MLR	
	R2	RMSEC	R2	RMSEC	R2	RMSEC	
CP (%)	0.95	0.74	0.86	1.63	0.85	5.74	
ADF (%)	0.93	1.25	0.84	4.96	0.70	8.91	
NDF (%)	0.94	1.31	0.85	3.60	0.72	4.59	
Notes.

CP crude protein

NDF neutral detergent fiber

ADF acid detergent fiber

R2 the coefficient of determination for the calibration set

RMSEC the root mean square error of calibration

The results of the cross validation are presented in Table 4. It can be seen in the table that the smallest value for prediction error (RMSEP value for CP, NDF and ADF is 1.24, 1.41 and 1.37, respectively) with the highest the coefficient of determination for the validation set (r2 value for CP, NDF and ADF is 0.91, 0.90 and 0.90, respectively) was provided by the PLS method. The results were less accurate for CP (RMSEP = 2.63, r2 = 0.86), NDF (RMSEP = 4.50, r2 = 0.82) and ADF (RMSEP = 5.86, r2 = 0.82) provided by the PCR method. The MLR had the worst prediction for CP (RMSEP = 6.74, r2 = 0.74), NDF (RMSEP = 8.59, r2 = 0.62) and ADF (RMSEP = 9.91, r2 = 0.60). Therefore, predictions were moderately successful for CP, NDF, and ADF was provided by the PLS method. The application of PCR using the whole wavelength region requiring scanning type spectrometers as with PLS, resulted in an almost twice as high prediction error. The same order of magnitude could be obtained for RMSEP value by using MLR. Compared to PCR and PLS, the MLR model uses only two characteristic wavelengths for the calculation, creating the conditions for constructing cheaper single-purpose filter instruments.

Table 4 Comparison of the accuracy of the validation results (n = 51) achieved by NIRS using three different methods for evaluation.

	PLS	PCR	MLR	
	r 2	RMSEP	RPD	r 2	RMSEP	RPD	r 2	RMSEP	RPD	
CP (%)	0.91	1.24	3.2	0.86	2.63	1.3	0.74	6.74	0.4	
ADF (%)	0.89	1.37	3.1	0.82	5.86	1.2	0.60	9.91	1.1	
NDF (%)	0.88	1.41	3.0	0.82	4.50	1.0	0.62	8.59	0.9	
Notes.

CP crude protein; NDF, neutral detergent fiber

ADF acid detergent fiber

r2 the coefficient of determination for the validation set

RMSEP room mean squared error of prediction

RPD the ratio of the standard deviation in the validation set over the room mean squared error of prediction

Similarly as RPD determined by the above three methods in Table 4, the accuracy was considered good for RPD > 2, acceptable for 1.4 < RPD < 2, and unreliable for RPD < 1.4 according to Albrecht et al. (2009) and Chang et al. (2001). Therefore, the highest RPD was provided by the PLS method for CP(RPD = 3.2), NDF(RPD = 3.0), and ADF(RPD = 3.1) while the worst RPD provided by the PCR and MLR method are also shown in Table 4.

Discussion

Samples selected for calibration (Table 1) comprised a wide range from 6.20 to 14.33% for crude protein (CP), from 50.71 to 71.08% for neutral detergent fiber (NDF) and from 35.13 to 42.34 for acid detergent fiber (ADF). This is confirmed by a certain instance in our study as an obvious difference among samples from the different distributing sites is the wide variability of sheepgrass. Hence, obtaining good relationship was more difficult for sheepgrass employed in this study.

Because the near-infrared spectrum contains all strength information of the chemical bond, chemical composition, electronegativity, etc. Meanwhile, other interference information, such as scattering, diffusion, special reflection, surface gloss, refractive index, and reflected light polarization, affects the near-infrared spectrum. Thus, to eliminate this effects of the later, there are three types of spectral normalization method: min/max normalization, vector normalization and zero correction. One common method such as the vector normalization was applied. Meanwhile, to restrain invariable background signals and to improve the visual resolution, the Savitsky-Golay 2nd order derivations were also applied in this study.

PLS regression analysis of the spectral data with CP content resulted in the highest coefficient of determination (R2) compared with PCR and MLR method (Table 3). Also, differences in standard error of full cross-validation (RMSEC) for CP prediction show the RMSEC achieved by PLS (0.74%) was more than 20% lower compared with PCR (1.63%) or MLR methods (5.74%). The RMSEC for CP of 0.74 gained by PLS is comparable with the range of 0.4–1.41 reported by Shi & Zhang (2011). There were not about PCR and MLR methods, respectively. The two wavelengths chosen for calibration by the MLR method were 1,366 and 1,418 nm for CP; 1,142 and 1,380 nm for NDF, and 1,148 and 1,566 for ADF. They are not within the range of 2,100–2,164 quoted by Redshaw et al. (1986) and Murray (1986). The MLR evaluation method for CP indicates a strong relationship (R2 of 0.85) with spectroscopic data, which suggests that this trait may be also accurately estimated by inexpensive filter instruments. Several authors compared results obtained by MLR and PLS in different products and concluded PLS and MLR give nearly the same prediction errors (Windham & Flinn, 1992; Rosental & Wlliams, 1996). In our study, PLS was more accurate than MLR method. One of the reasons might be that 9 components were used for PLS compared with 5 PC used for PCR and 2 wavelengths used for MLR.

In our paper the RMSEP values were all higher than RMSEC results, similar to the ones reported by Stimson et al. (1991). The lowest RMSEP achieved for CP content among the methods applied may be due to the fact that CP content was recalculated to the dry weight while CP was expressed on DM basis. Differences between RMSEC and RMSEP values may be due to the limited number of samples. This clearly indicates the problem of obtaining representative samples in practice. Also, sheepgrass samples, which are very heterogeneous, were scanned with pretreatment while for wet chemistry analysis the same samples were dried and ground. Although errors are often slightly higher for samples scanned in their natural state than for dried and milled samples, this is balanced by the ability to scan much larger samples, the avoidance of compositional losses and changes due to oven drying and a major reduction in analysis time and cost due to no sample preparation being necessary (Wilman et al., 2000).

The findings indicate spectroscopic data evaluated by PLS method were strongly related to reference values and had lower RMSEC values and highest RPD compared with PCR and MLR methods. By studying the Figs. 3, 4 and 5, the same conclusion can be drawn as in case of the prediction of CP, ADF and NDF content by PLS with the highest correlations of determination (r2) for the validation set (n = 51). Our results in this study indicated for the first time that the quality of sheepgrass hay could be successfully evaluated by the NIRs with PLS regression method.

Figure 3 Relationships between the measured and predicted values of the crude protein content (CP) of sheepgrass hay for the validation data set.

The red line represents the best fit.

Figure 4 Relationships between the measured and predicted values of the acid detergent fiber content (ADF) of sheepgrass hay for the validation data set.

The red line represents the best fit.

Figure 5 Relationships between the measured and predicted values of the neutral detergent fibre content (NDF) of sheepgrass hay for the validation data set.

The red line represents the best fit.

Conclusion

The NIRs prediction of forage chemical value is a relatively inexpensive, rapid and reliable method compared with reference methods, requiring a relatively small quantity of sample and predicts several concentrations of components simultaneously. An important advantage of NIRs is its ability to analyze samples without chemical treatment, hence costs and chemical wastes can be reduced by using accurate NIRs model. The success of NIRs analysis depends almost entirely on the reliability of the primary reference data used at calibration. Using NIRs can predict CP, NDF, and ADF contents in sheepgrass hay samples without costly or lengthy pretreatment as shown in this paper. Nevertheless, a satisfactory accuracy with an average standard error of prediction by PLS method of 0.74, 1.31 and 1.25 for CP, NDF and ADF, respectively, was obtained. The comparison of validation statistics (r2 and RMSEC) among PLS, PCR and MLR equations showed PLS to be the most accurate (However, it does use 9 factors compared with 5 for PCA and 2 for MLR.). The MLR evaluation method for CP has the potential to be used for industry requirements, as it needs less sophisticated and cheaper instrumentation using only a few wavelengths.

Supplemental Information

Appendix S1 The information of sample sites

Click here for additional data file.

Appendix S2 The chemical value and NIR spectra of samples

Click here for additional data file.

The authors thank Prof. Wu Jingzhu, Li Lujun and Yang Gaowen for their assistance in modification, and Liu Qiqi and Wu Shuheng for the help with the laboratory-based NIR measurements.

Additional Information and Declarations

Competing Interests

Author Contributions

Data Availability

The authors declare there are no competing interests.

Jishan Chen conceived and designed the experiments, performed the experiments, analyzed the data, wrote the paper, prepared figures and/or tables.

Ruifen Zhu performed the experiments, analyzed the data, wrote the paper, prepared figures and/or tables.

Ruixuan Xu contributed reagents/materials/analysis tools, reviewed drafts of the paper.

Wenjun Zhang reviewed drafts of the paper.

Yue Shen contributed reagents/materials/analysis tools.

Yingjun Zhang conceived and designed the experiments, analyzed the data.

The following information was supplied regarding data availability:

The research in this article did not generate raw data.

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
