# Peer review of "Evaluation of Leymus chinensis quality using near-infrared reflectance spectroscopy with three different statistical analyses"

_PeerJ, doi:10.7717/peerj.1416_

## Round 0.1 · original submission · Minor Revisions

Dear Authors,

Your manuscript has been reviewed by three independent reviewer and all expressed overall satisfactions on the manuscript (their comments are attached). However, there are some minor points that need to be taken care of. Also some additional editing is required. I encourage you to complete these tasks and submit the revised version soon.

With best regards.

·

Basic reporting

The ms is well written, don’t have any problem with the English. Tables and figures are adequate and relevant. The authors explain step to step the procedure carried out in the experiment: hay sampling, mathematical procedure, analystical methds, etc. The ms is easy to understand and read. In definite, it is a good ms.

Experimental design

The experimental design is adequate with the objectives planted in this experiment.
I want to stand out in the experiment that. I) the number of samples used in this assay give a high degree realistic to the assay; ii) there was wide range of values in the parameters CP, ADF, and NDF what it is also important to obtain a good calibration and prediccition, and iii) it is important to use several mathematical model to know what method is the better in the predccidions.

Validity of the findings

The validation was done in good way. The big number and variability in the samples did that the authors obtained a good calibration and let them to choose the best model to predict CP, ADF and NDF parameters.

Additional comments

This is a good ms, well written and discussed. With an objective novels, where the use of NIR is being development in a lot of agricultural areas. This method avoid complicated chemical analysis, it is cheap, fast, and reliable.

·

Basic reporting

this manuscript is well written and tried to evaluate of Leymus chinensis quality using Near-Infrared reflectance spectroscopy, as well as the effect of three different statistical analyses on the chemical quality of sheepgrass. The scope is important and fit to this journal. The manuscript has some formatting errors (e.g. line 72: 117.19 ten thousand tons).

Experimental design

The manuscript describes the NIRS prediction of quality factors in sheepgrass. The work has been carried out in the correct manner and is of reasonable scientific interest.

Validity of the findings

As an important perennial forage grass across the Eurasian Steppe, sheepgrass is known for its adaptability to various environmental conditions, high palatability, rich in nutrients and all kinds of livestock feeding. Hence, sheepgrass in this region is regarded as an important productive grass for the hay industry developing rapidly due to a prosperous status in the dairy industry. However, it remains unclear whether NIRs is applied to search the valuable information about NIRs prediction models and rapidly and accurately assess the quality of the hay dominanted by sheepgrass for CP, ADF and NDF.therefor, this work is very important for sheepgrass.

Additional comments

formatting errors could be improved.

Reviewer 3 ·

Basic reporting

Background information was sufficient. No major issue with the language.
Relevant figures, manuscript complied with basic reporting structure.

Experimental design

Good

Validity of the findings

Statistical analysis and comparison was good. For NIR data, normalization procedure across various samples needs to be discussed.

Additional comments

Relevant and may be useful for industry

---

## Round 0.2 · accepted · Accept

I am happy to inform you that your manuscript has been reviewed by three independent reviewers and all of them have recommended it in favor of acceptance.

·

Basic reporting

The authors have corrected property the suggestions done for the reviewers. And now the ms can be published in this Journal. In my previous report my evaluation was very positive for this ms for the topic of the ms, where the experimental desing and the mathematical procedure were adequate.

Experimental design

In my previous report my evaluation was very positive in this section.

Validity of the findings

In my previous report my evaluation was very positive in this section.

Additional comments

The authors have corrected property the suggestions done for the reviewers. And now the ms can be published in this Journal. In my previous report my evaluation was very positive for this ms for the topic of the ms, where the experimental desing and the mathematical procedure were adequate.

·

Basic reporting

Very well and accept to publication

Experimental design

Good

Validity of the findings

Well and acceptable

Additional comments

Good

Reviewer 3 ·

Basic reporting

reviewed appropriately

Experimental design

Addressed appropriately

Validity of the findings

Addressed appropriately

Additional comments

NA